# Neuron tracing and quantitative analyses of dendritic architecture reveal symmetrical three-way-junctions and phenotypes of *git-1* in *C. elegans*

Omer Yuval[1,2], Yael Iosilevskii[1], Anna Meledin[1], Benjamin Podbilewicz[1]*, Tom Shemesh[1]*

1 Faculty of Biology, Technion–Israel Institute of Technology, Haifa, Israel, 2 School of Computing, Faculty of Engineering and Physical Sciences, University of Leeds, Leeds, United Kingdom

* podbilew@technion.ac.il (BP); tomsh@technion.ac.il (TS)

**Data Availability Statement:** The authors confirm that all data underlying the findings are fully available without restriction. All relevant data are within the paper and its Supporting Information

## Abstract

Complex dendritic trees are a distinctive feature of neurons. Alterations to dendritic morphology are associated with developmental, behavioral and neurodegenerative changes. The highly-arborized PVD neuron of *C. elegans* serves as a model to study dendritic patterning; however, quantitative, objective and automated analyses of PVD morphology are missing. Here, we present a method for neuronal feature extraction, based on deep-learning and fitting algorithms. The extracted neuronal architecture is represented by a database of structural elements for abstracted analysis. We obtain excellent automatic tracing of PVD trees and uncover that dendritic junctions are unevenly distributed. Surprisingly, these junctions are three-way-symmetrical on average, while dendritic processes are arranged orthogonally. We quantify the effect of mutation in *git-1*, a regulator of dendritic spine formation, on PVD morphology and discover a localized reduction in junctions. Our findings shed new light on PVD architecture, demonstrating the effectiveness of our objective analyses of dendritic morphology and suggest molecular control mechanisms.

## Author summary

Nerve cells (neurons) collect input signals via branched cellular projections called dendrites. A major aspect of the study of neurons, dating back over a century, involves the characterization of neuronal shapes and of their dendritic processes.

Here, we present an algorithmic approach for detection and classification of the tree-like dendrites of the PVD neuron in *C. elegans* worms. A key feature of our approach is to represent dendritic trees by a set of fundamental shapes, such as junctions and linear elements. By analyzing this dataset, we discovered several novel structural features. We have found that the junctions connecting branched dendrites have a three-way-symmetry, although the dendrites are arranged in a crosshatch pattern, and that the distribution of junctions varies across distinct sub-classes of the PVD's dendritic tree. We further quantified subtle morphological effects due to mutation in the *git-1* gene, a known regulator of

files. Computer code available at: https://github.
com/Omer1Yuval1/Neuronalyzer Microscopy
images, videos and figure data available at: https://
gin.g-node.org/OmerYuval/Neuronalyzer_SI.

**Funding:** This work was supported by the Israel
Science Foundation (grant 2751/20 to TS and 257/
17 to BP). We thank the *C. elegans* knockout
consortium for the git-1(ok1848) deletion allele
generated by the C. elegans Gene Knockout Project
at the Oklahoma Medical Research Foundation, and
git-1(tm1962), generated by the National
Bioresource Project, Tokyo, Japan. Some strains
were provided by the Caenorhabditis Genetics
Center, which is funded by National Institutes of
Health Office of Research Infrastructure Programs
(P40 OD010440). The funders had no role in study
design, data collection and analysis, decision to
publish, or preparation of the manuscript.

**Competing interests:** The authors have declared
that no competing interests exist.

dendritic spines. Our findings suggest molecular mechanisms for dendritic shape regula-
tion and may help direct new avenues of research.

## Introduction

The link between the information processing function of neurons and the intricate shapes of
their dendritic processes has been the subject of over a century of extensive study [1–3]. Mor-
phological features of dendritic arbors, such as length, diameter, and orientation have been
related to neuronal functions [4–9], while deformations of the dendritic architecture have
been associated with diseases and disorders [10–13]. Dendritic morphology varies significantly
across species and across neuronal classes, which are often associated with distinct characteris-
tic shapes [1–3,14–16].

Over the past decade, the *C. elegans* bilateral PVD neuron has become a powerful model to
study dendritic patterning, owing greatly to its stereotypical, highly ordered structure in the
fourth larval stage and young adult (Fig 1A) [17]. The PVD neuron is located between the
hypodermis and its basement membrane [17,18], and functions as a polymodal nociceptor,
mechano- and thermo-sensor, and was suggested to play a role in proprioception–the sensing
of body posture [18,19]. Two PVD cell bodies are born post-embryonically in the second larval
stage (L2) and develop across the last three larval stages before adulthood (L2-L4), through a
dynamic process of dendritic growth and retraction. By late L4 (typically within ~30 hours at
20˚C), their complex dendritic arbor of repeating candelabra-shaped units extends almost the
entire surface area of the worm [17,18]. This highly organized structure is made of tubular pro-
cesses with diameters ranging between 35–60 nm [17,18,20]. Studies have implicated the PVD
in a variety of functions, including self-fusion, aging and regeneration, and more recently for
exploring structural aspects of neuropsychiatric and neurodegenerative diseases [17,21–25].
The relatively complex and high-order arborization of the PVD makes it an excellent model
for studying the connection between structure and function in the nervous system
[9,17,18,26–28].

A known regulator of dendritic morphogenesis and synaptic plasticity is GIT1: a GTPase
activating protein for the small GTPase Arf (ADP ribosylation factor), as well as a focal adhe-
sion scaffolding protein. Mutations in GIT1 have been linked with fear response and learning
impairment, as well as reduced motor coordination and altered walking gait in rodents
[29,30]. It has also been implicated in Huntington's disease, Schizophrenia, and Attention Def-
icit and Hyperactivity Disorder (ADHD) [31–34]. Moreover, PIX (p21-activated kinase
[PAK]-interacting exchange factor), the most prominent binding partner of GIT1, has been
implicated in mental retardation in humans [35]. Several studies in rodent models have identi-
fied a link between GIT1 mutation and a reduction in the number of dendritic spines
[29,36,37] and a recent report suggests it may have a structural role in the PVD as well [38].

Analysis of neuron architecture faces two fundamental challenges; foremost of which is the
extraction of cellular features from microscopy images [39]. Due to the inherent noise associ-
ated with optical microscopy of living systems, identifying and isolating the relevant image
portions is a technically difficult and error-prone process [40–42]. Following segmentation,
there remains the second challenge of interpreting and analyzing the segmented structures. As
neurons display elaborate and varied shapes, characterization of their morphologies requires
complex metrics and classifications, often developed or adjusted for a specific cell type
[8,9,43]. In particular, the dendritic arbors of PVD neurons are described in terms of visually-
striking, repetitive morphological elements, referred to as 'menorahs' or multibranched

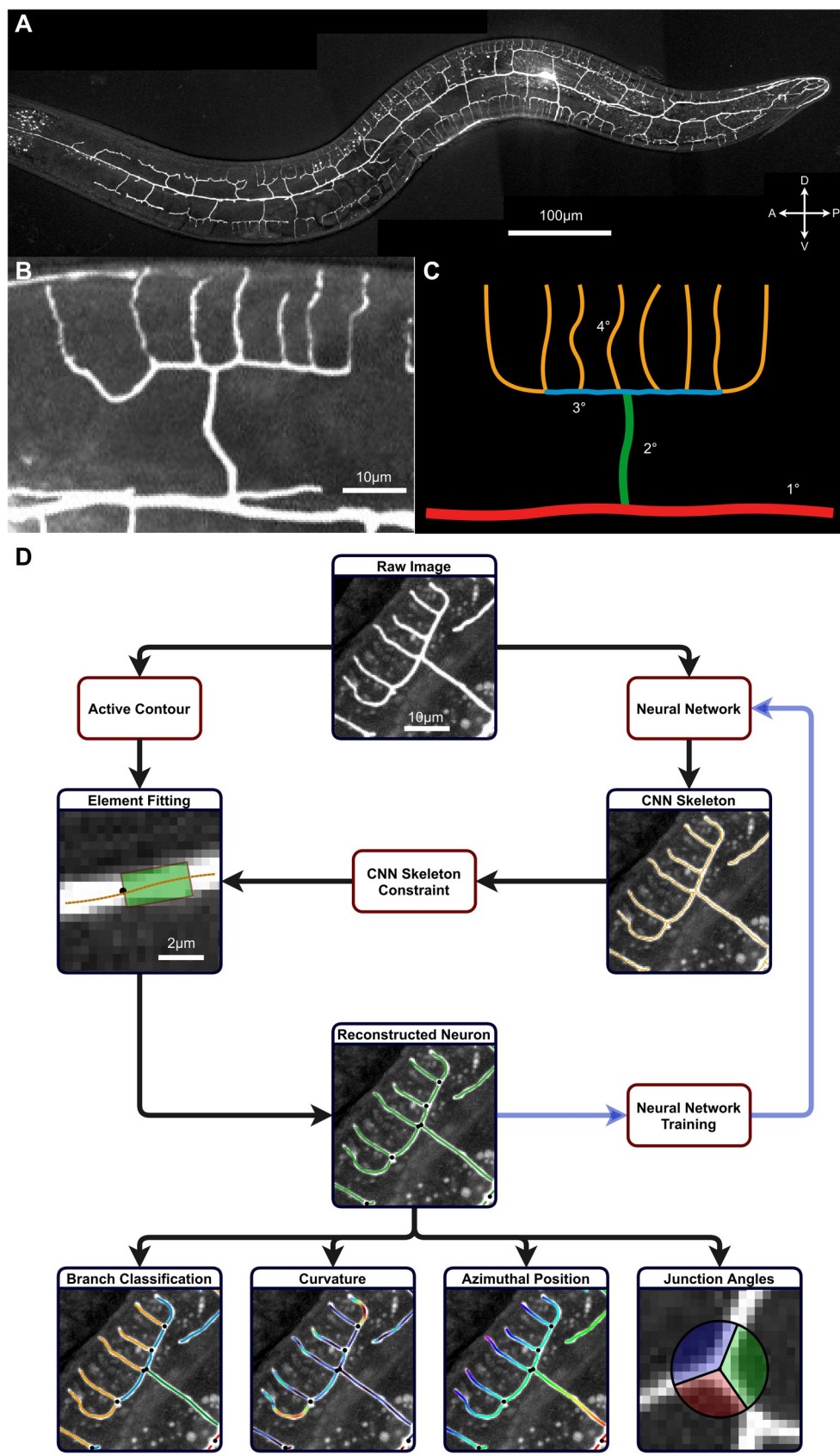

**Fig 1. Tracing and feature extraction of the PVD neuron. A.** A maximum projection image of the PVD neuron in a young-adult wild-type *C. elegans* expressing fluorescent Kaede. A = anterior, P = posterior, D = dorsal, V = ventral. **B.** A characteristic 'menorah' structure, a repetitive pattern in the PVD's dendritic arbor. **C.** A diagram of the menorah showing the conventional classification into branch orders. **D.** Outline of the algorithm for tracing and analysis of the PVD neuron. Starting with a grayscale raw image, dendritic processes are traced by fitting them with rectangular masks. This process is assisted by a skeleton, derived from a pre-trained convolutional neural network (CNN) used to constrain the orientation of the fitted elements. This results in accurate reconstruction of the PVD shape, that may be added to the CNN training set in order to refine its output for future analysis. The resulting neuron reconstruction can then be used to extract and quantify morphological features.

candelabra [17,26,44] (Fig 1B and 1C). Characterization of the PVD dendritic morphology is therefore accomplished by annotation and quantification of the menorah structure (Fig 1C). Due to the intricacy of these challenges, both extraction and analysis of PVD morphology are usually performed semi-manually, using time-consuming methods that prohibit large-scale studies. In addition, human characterization of neuronal shapes is highly subjective and often not sensitive enough for subtle, small-scale structures, raising the need for a high-precision, unbiased, quantitative analysis of morphological features [17,21,44,45]. While multiple image-processing tools are utilized to assist in tracing of neuronal elements [40–42,46], most methods do not easily lend themselves to post-tracing quantitative analysis. Moreover, general-use segmentation algorithms do not consider how the geometry of the neuronal processes is related to the overall shape of the *C. elegans* organism. An integrated tool that combines detection, segmentation and analysis of the PVD's dendritic tree is therefore required.

Here, we present a computational tool for the automated segmentation and analysis of PVD neuronal architecture. Tracing neuronal processes from microscopy data is performed by a hybrid algorithm, combining a region-based active-contour model for feature extraction [47–49], with a deep learning network for image filtering [50,51] (Fig 1D). While the neuron tracing functions may be assisted by a human operator, segmentation operates in a fully autonomous mode and batch-processes multiple raw microscopy images. Following the tracing step, quantification and classification of the extracted features are performed algorithmically (Fig 1D). Finally, the tracing and analysis outputs are catalogued, stored and shared in a neuronal morphology database. We utilize the collected morphological data for two tasks: first, by analyzing an extensive set of quantified PVD structures, we aim to update the current histological knowledge of the PVD neuron. This detailed knowledge can afford insight into the underlying mechanisms that sculpt neuronal structures. Further, the morphological database is used to quantify the effect of *git-1* mutation on the *C. elegans* PVD neuron during early adulthood. Thus, also demonstrating the use of the tool for the detection and quantification of subtle morphological differences as a result of mutations that were implicated in human disease.

## Results

### Extraction of PVD morphology from microscopy data

Deep-learning algorithms are widely used for segmentation of microscopy data in general, and neuronal shapes in particular [51,52]. Here, we designed, trained and applied a convolutional neural network (CNN; S1 Fig) in order to extract the neuron signal from images of one-day adult *C. elegans* (Fig 1A). The CNN acts to classify the fluorescent signal as either attributed to the PVD or to background noise and non-neuronal labeled components such as auto-fluorescent gut granules [53]. The CNN operates on small sub-regions of the image (see Fig 2A for an ensemble of raw and labeled regions), which are then re-assembled to reconstruct the full neuron image. The CNN outputs an image-sized array with values corresponding to the classification of each image pixel into PVD and non-PVD elements. Next, the CNN output is

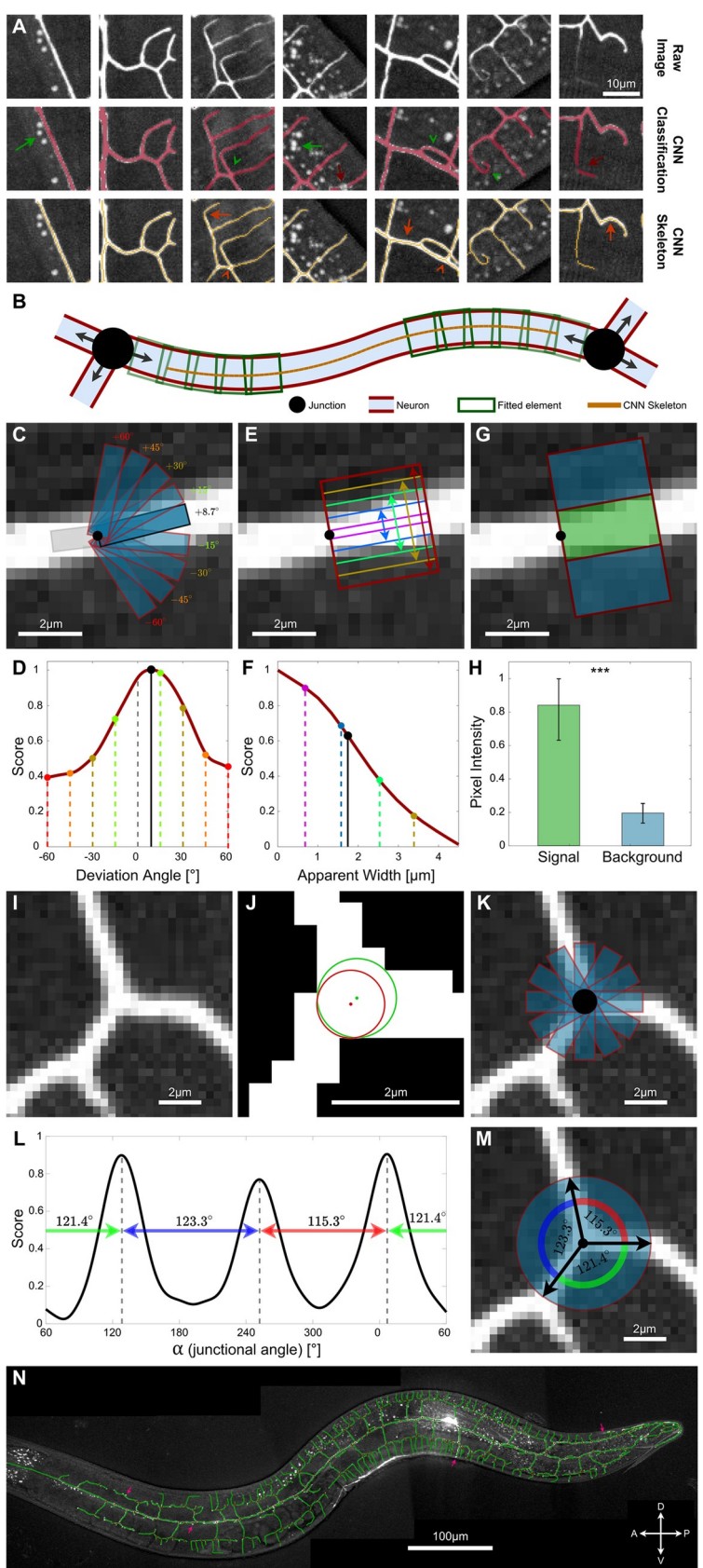

**Fig 2. Segmentation of the PVD using neural networks and active-contour models. A.** A convolutional neural network (CNN) is trained to detect PVD elements in noisy image patches, and to classify neuron and non-neuron pixels. The network gets a small image patch (top row, "Raw Image"), and returns a classified image array of the same size (middle row, "CNN Classification"). Green arrows show examples of correct classification of high-intensity autofluorescent gut granules, green arrowheads show examples of correct classification of complex morphologies, and red arrows show examples of incorrect classification of neuron pixels. The CNN output image is subsequently reduced to a topological skeleton (bottom row, "CNN Skeleton"). Orange arrows and arrowheads indicate cases in which the skeleton does not accurately capture the geometry of the neuron. These cases include over- and under-smoothing of neuronal processes (orange arrows), and deformed junction geometries (orange arrowheads). **B.** An illustration of the tracing process of a single neuronal segment. Sequential convolution of rectangles (green) is applied from the endpoints inwards until the segment is fully traced. **C, D.** The local orientation of a PVD element (C) is determined by the optimal alignment of a rectangular mask (blue) with the dendritic process (white), corresponding to the maximal convolution score (D, dashed lines represent detected deviation angle). **E, F.** Apparent local width of a PVD element (E) found using the decay rate (second derivative) of the convolution function (F, dashed line represents detected apparent width), using the orientation found in (C-D). **G, H.** The intensity and variation of the local background are determined by sampling the pixels within rectangles on both sides of the one found in C-F (G, blue rectangles for the background and green for the dendritic process). These are used to normalize the score and account for variation in background throughout the image and across images. Statistics were calculated using the nonparametric Mann–Whitney test. ***$p < 0.0005$. Error bars show the standard deviation. **I-M.** Determining the geometry of neuronal junctions–center point, radius, and angles. Given an approximate center point from the skeleton image (red dot in J), the precise center is detected by optimally fitting a circular mask to the binarized CNN image (green dot in J). Following the detection of the precise center and radius (J), a radially aligned rectangular mask (blue) is convolved against the greyscale image (K). Peaks in the convolution function (L) correspond to the detected local junctional angles (α; M). **N.** An example of a fully traced wild-type PVD neuron (same neuron as Fig 1A). Pink arrows indicate completely or partially untraced segments.

skeletonized, producing a pixel-wide, thin representation of the PVD shape (Figs 1D and S2C). While the neural network is highly efficient in classification of microscopy images, we found that the CNN-derived shape is insufficient for a faithful representation of the PVD (see S2D Fig).

In order to accurately extract the PVD morphology, we employed a simplified region-based active contour model to segment raw microscopy images [47–49]. Briefly, active contour methods detect morphological features by optimally fitting predefined shapes to the data. The optimization is influenced by "external forces" that attract the model shape to specific image elements, and "internal forces" that enforce intrinsic properties of the model shape, such as curvature [54]. Since the PVD dendrites have a tubular morphology with a slowly varying diameter, we represented the PVD shape by a series of discrete rectangular elements (Fig 2B; green rectangles). The fit score of each rectangular element (corresponding to the "external energy" of the active contour) is defined as the convolution of a rectangular mask with the image data. The score ranges from 0, corresponding to a mask containing no PVD elements (background, blue; Fig 2G and 2H), to 1 for a mask containing only PVD elements (signal, green; Fig 2G and 2H). The orientation of the dendritic process at each location is found by considering possible deviations from the orientation of the preceding element, where a 0˚ deviation indicates a locally linear segment (Fig 2C). The optimal orientation is defined as maximizing the mask convolution score (Fig 2C and 2D). The width of the rectangular mask is fitted at each point along the dendritic process by calculating the characteristic length at which the fit score decays with the rectangle width (Fig 2E and 2F; see arrows). Note that the fitted rectangular width does not indicate the actual diameter of the dendritic process, which falls below the resolving power of the microscopy setup, but rather the apparent fluorescent signal of the processes, which varies significantly throughout the PVD [17]. The length of the rectangular elements was taken as a twice the rectangular width (see S1 Table). Once a section of the PVD has been fitted with a mask, the local background intensity in the vicinity of the dendrite is calculated and subtracted from the image data (Fig 2G and 2H). Thus, detection of dendritic processes is assisted by the segmentation of neighboring regions. Ultimately, the shape of the

dendritic processes is represented by a connected chain of rectangles (Fig 2B). We regulated the continuity and smoothness of the representative chain ("internal energy" of the active contour) by constraining the orientations and widths of successive rectangles (see S1 Text).

In addition to linear elements, the PVD arbors are characterized by junctions that connect dendritic branches (Fig 2I). We represented such junctions as circular elements, from which rectangular processes emanate radially. First, the position and size of the junctions are found by optimally fitting a circular mask (Fig 2J). Next, the dendritic processes that originate from each junction are detected by calculating the convolution score of rectangular masks across a range of orientations around the junction center (Fig 2K). The maxima of the score for different orientations correspond to the set of junctional angles, $\{\alpha_1, \alpha_2, \alpha_3\}$, that characterize the junction's shape (Fig 2L and 2M).

Sequential active contour segmentation is highly sensitive to the choice of the starting point, as well as image noise [47,55]. Since the active contour fitting was performed on the unclassified, original microscopy images, we overcame this limitation by utilizing the information contained in the CNN-derived skeleton to initialize and guide the active contour process. First, tips and vertices of the CNN skeleton are detected (pixels connected to either one, or more than two pixels, respectively). Skeletonized vertices serve as markers for dendritic junctions, and are used as the initial guess positions for fitting their shapes. Moreover, during the active contour tracing of the linear dendritic processes (Fig 2B–2H) we require the rectangular tracing elements to contain pixels of the CNN-derived skeleton from the corresponding skeleton segment. This constraint combines the benefits of deep learning and active contour fitting, and prevents divergence of the tracing into image regions that were classified as non-PVD by the CNN. Finally, we utilized the output of the tracing algorithm, as well as manual corrections, to update and retrain the CNN (see Fig 1D and S1 Text). This feedback loop between the deep-learning classification module and the model-based fitting module is designed to enable a robust and adaptive extraction of the PVD neuron morphology from noisy images.

The combined application of these segmentation procedures results in a dataset containing the position, dimensions, orientation and connectivity of the rectangular elements. Visual inspection confirms that this data serves as an efficient and faithful representation of the PVD shape (*e.g.* Fig 2N). The abstracted nature of this representation facilitates post-segmentation analysis and may be used to combine multiple images to form larger datasets.

## Analysis of PVD shape

Following feature detection, we turned to construct a quantified characterization of the PVD architecture. In order to relate the structure of the dendritic arbor to the shape of the organism, we first defined an appropriate global system of coordinates. Subsequently, the extracted dataset of morphological elements was used to identify spatial patterns that represent intrinsic motifs of PVD organization. Classification of PVD elements according to such motifs is used to provide an idealized PVD representation, as well as a basis for quantitative comparison between phenotypes.

### PVD coordinate system

It is convenient to describe the position of the extracted rectangular PVD elements by the natural coordinate system for *C. elegans*, defined by the moving frame $(\hat{S}, \hat{D}, \hat{R})$; where $\hat{S}$ is locally tangential to the worm's longitudinal axis, $\hat{D}$ points along the local dorsal direction, and $\hat{R}$ completes the orthonormal set by pointing in the direction of the worm's right (see Fig 3A). The worm's longitudinal axis is detected by morphological dilation and erosion image operations (see Figs 3B, S3A and S3B and S1 Text for details).

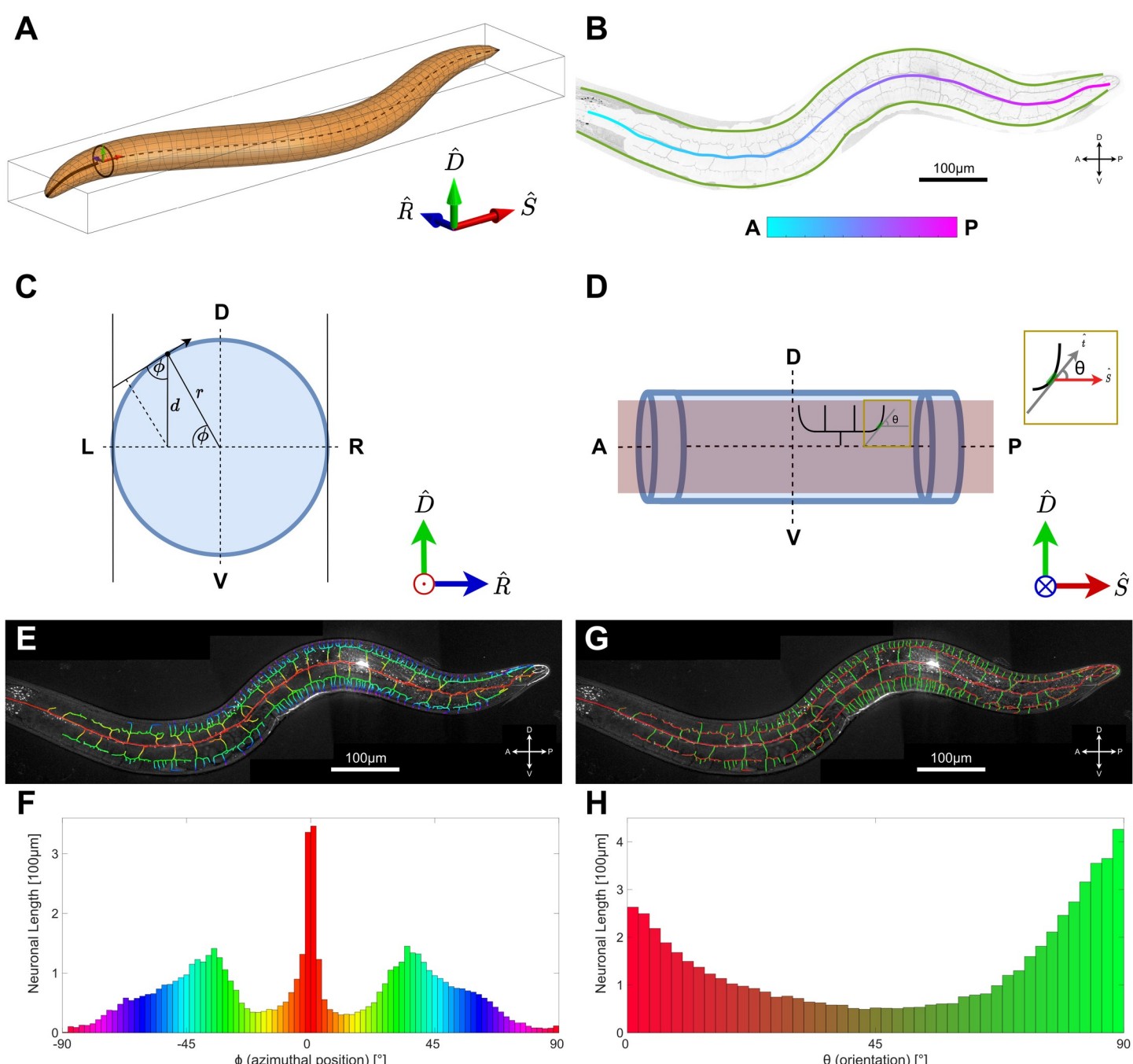

**Fig 3. The worm's coordinate system and PVD feature extraction. A.** The coordinate system used to characterize the PVD neuron is defined by $(\hat{S}, \hat{D}, \hat{R})$; where $\hat{S}$ is locally tangential to the worm's longitudinal axis, $\hat{D}$ points along the local dorsal direction, and $\hat{R}$ points in the direction of the worm's right. **B.** Using the projection image of the non-planar PVD neuron, an outline image is generated using morphological operations applied to the neuron's trace. This is used to find the neuron's centerline and borderline. **C.** A schematic of the worm's cross section. The azimuthal angle $\phi$ denotes the azimuthal position of PVD elements. r, radius at each point; d, distance. **D.** A schematic of the worm from a left/right side view. The angle $\theta$ denotes the orientation of PVD elements, defined as the angle between the longitudinal axis, $\hat{S}$, and the local tangent, $\hat{t}$. **E, F.** Visualization of the azimuthal angle $\phi$ (E) and distribution of the PVD elements for different $\phi$ angles (F, n = 10). The peaks at 0 and $\pm 35°$ correspond to the primary (red) and tertiary (green, dorsal are positive and ventral are negative values) branches. Same neuron as Fig 1A. **G, H.** Visualization (G) and distribution (H) of orientation angle, $\theta$, for PVD elements (n = 10). The distribution shows that most of the neuron's length is either parallel (red) or perpendicular (green) to the midline. D = dorsal, V = ventral, L = left, R = right, A = anterior, P = posterior.

Since the PVD lies close to the worm's body surface, positions of its dendritic elements may be given by only two coordinates: distance $s$ along the longitudinal axis (defined as positive posteriorly towards the tail, with the origin anteriorly at the head; A-P in Figs 3B and S3B), and the azimuth angle $\phi$ relative to the $\hat{R}$ axis (defined as positive for a counter-clockwise rotation about $\hat{S}$, such that azimuth is positive for dorsal elements, and negative for ventral; see Fig 3C). Note that two PVD neurons cover the worm laterally (PVDL, PVDR); this analysis addresses the PVD as viewed from the left. PVDR images were flipped to provide the same frame of reference. Although the worm's body surface is curved, microscopy images are projected to the flat image plane defined by $\hat{S}$ and $\hat{D}$ vectors. Coordinates of an element in the image plane are therefore given by the distances $(s, d)$, and the transformation between image and worm coordinates is given by $\phi = sin^{-1}(d/r(s))$, where $r(s)$ is the radius of the worm at each point along the longitudinal midline axis. For each element of the dendrites, we defined the midline orientation angle $\theta$ of the local tangent, $\hat{t}$, relative to the longitudinal axis: $\theta = tan^{-1}(dD/dS)$, where $dD$ and $dS$ are length components in the $\hat{D}$ and $\hat{S}$ directions respectively (see Fig 3D).

## Classification of PVD processes

Based on these definitions, we characterized the spatial distribution of PVD dendrites. We found that PVD elements are spread non-homogeneously in the azimuthal direction, and are predominantly localized at azimuths $\phi \approx 0°$ and $\phi \approx \pm 35°$ (Fig 3E and 3F). The distribution of dendritic processes was found to be symmetrical about the longitudinal midline. As worms were oriented on their sides in all images, the peak at an azimuth of $\phi \approx 0°$ corresponds to elements of the PVD's primary branch, while elements belonging to secondary, tertiary and quaternary branch orders are located at progressively increasing $|\phi|$ (Fig 3E and 3F). Visual inspection of color-coded PVD segments in specific worms further indicates a correspondence between azimuthal angles and branch orders (Fig 3E). Next, we examined the local orientation of dendritic processes, $\theta$. Our analysis showed that the distribution of the orientation angles is heterogenous throughout the PVD, yet is roughly symmetric with respect to the longitudinal axis (Fig 3G and 3H). We found that dendrites are most likely to locally orient either parallel to the worm axis ($\theta = 0°$; red), or normal to it ($\theta = 90°$; green) (Fig 3H). Visual inspection of the orientational distribution showed that PVD elements form a network with rectangular appearance, where connected segments are mostly perpendicular to each other (Fig 3G and 3H).

Following these observations, we tested whether there is a correlation of local midline orientation, $\theta$, of dendritic processes with their azimuthal position, $\phi$. By plotting the distribution of dendritic elements vs. both azimuth and orientation, we found that PVD elements are strikingly clustered into distinct classes (Fig 4A). As the clusters were roughly symmetric about the midline, we grouped together dorsal and ventral classes (corresponding to the same magnitudes of azimuthal angles). We determined the intrinsic azimuth and orientation ranges of these morphological classes by finding the threshold density in the $(\theta, \phi)$ space which results in four distinct clusters (Fig 4A). Subsequerntly, we classified complete PVD segments of individual worms according to the most abundant class of rectangular elements in each segment, where segments are defined as the set of linear elements connecting two neighbouring junctions/tip (see Fig 4B and 4C). We have found that the consensus classification of the segments closely corresponds to conventional manual classification of menorah orders by visual inspection (see Figs 1C and 4C) [17,18,21]. In order to relate our analysis to earlier works, we named the morphological classes according to the conventional menorah branch order terminology: class-1 (corresponding to primary branch), class-2 (secondary), class-3 (tertiary) and class-4

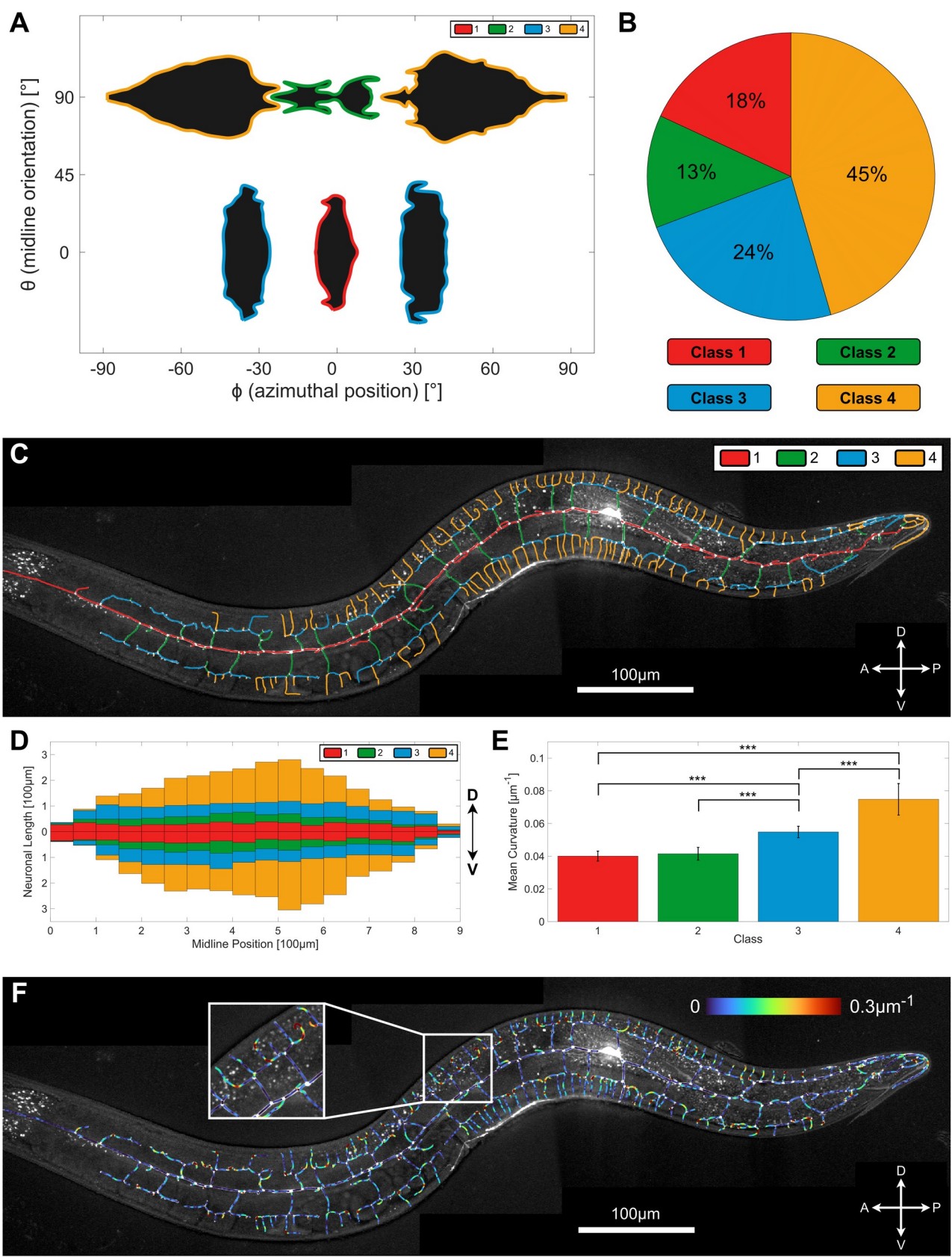

**Fig 4. Morphological characterization of the wild-type PVD. A.** Distribution of PVD elements by the azimuthal angle, $\phi$, and midline orientation angle $\theta$. Four distinct morphological classes are found: Class 1 (red), class 2 (green), class 3 (blue) and class 4 (yellow) (n = 10). **B.** The percentage of each of the morphological class found in (A); (n = 10). **C.** Visual example of the algorithmically-derived classification (same neuron as Fig 1A), closely resembling conventional branch-order classification scheme (see Fig 1C). **D.** Distribution of neuron length elements along the midline, classified as in C and averaged across worms (n = 10). **E.** Mean curvature of neuronal elements for each morphological class. Class 4 elements are the most curved, whereas class 3 elements are more curved than class 1 and 2, but less than class 4. The difference between class 2 and 4 is also statistically significant. Statistics were calculated using the nonparametric Mann–Whitney test. $^{***}p < 0.0005$. Error bars show the standard deviation (n = 10). Same neuron as Fig 1A. **F.** Example of a PVD neuron color-coded for curvature. Same neuron as Fig 1A.

(quaternary). Importantly, the algorithmically derived classes are similar, but not identical to the menorah ordering scheme described previously [17], (see S4A Fig). While noting this potential disparity, we found that the consistency and robustness of the algorithmic classification provide an unbiased and reproducible annotation of the PVD dendrites in the wild-type L4 or young adult (see S4B–S4J Fig for more examples). For an ideal candelabrum (as in Fig 1C) the orders and classes are usually synonymous. However, branches that differ from the idealized shape are typically labeled as ectopic by manual classification, while the algorithm classifies all branches as belonging to the most similar morphological class.

Using the derived classifications, we next quantified how the total length of PVD dendrites is allocated to the different morphological classes (Fig 4A). We found that almost half of the PVD processes length is made up of class-4 branches (Fig 4B and 4C; yellow). We further examined the relative contribution of different morphological classes at different positions along the worm's longitudinal (A-P) axis resulting in a schematic of a typical PVD morphology (Fig 4D). As expected, we found that class-1 processes are located near the longitudinal midline axis (red), and appear uniformly distributed along the head-tail axis. In agreement with previous works [21] PVD sections belonging to higher order classes become more prominent near the cell body, and taper towards its ends (Fig 4D).

Finally, we computed the curvature for all PVD elements, and compared the averaged value of the unsigned curvature for different classes. We found that the mean curvature of class-3 elements is significantly higher than class-1 and class-2, whereas class-4 elements are significantly more curved than classes 1–3 (Fig 4E). Visual inspection of a PVD color-coded for curvature confirms these results (Fig 4F). In particular, class-4 segments at the anterior and posterior ends of each mernoah often contain a highly curved region that gives rise to an L-shaped segment (Fig 4F). These results raise the possibility that by including additional metrics, such as line curvature, the classification of PVD elements may be further enhanced.

## Three-way junctions' angles are symmetric on average

Our analysis has shown that PVD junctions connect segments that are, on average, perpendicular to each other. In order to test whether this arragment results from forces that orient the dendritic processes or from intrisic properties of the dendritic junctions, we analyzed the geometry of individual junctions. We found that PVD processes are predominantly connected by 3-way junctions (98.3% of all junctions), while 4-way junctions are scarce (1.7% of all junctions), and higher order junctions were not detected. Next, we determined the relative orientations of the dendritic processes at the points of their connection to the junctions. We observed that each 3-way junction is characterised by an ordered triplet of angles $\{\alpha_1, \alpha_2, \alpha_3\}$, where $\alpha_1 \leq \alpha_2 \leq \alpha_3$ and $\alpha_1 + \alpha_2 + \alpha_3 = 360°$ (see Fig 5A for examples). Note that, in general, the junction angles differ from the orientations of the processes to which they are connected (see S5A Fig). After correcting for the distortion of the junction angles due to projection to the image plane (see Fig 3B and 3C and S1 Text for details), we found the averages for the smallest, intermediate and largest angles to be $\langle \alpha_1 \rangle = 92°$, $\langle \alpha_2 \rangle = 119°$, and $\langle \alpha_3 \rangle = 149°$ (Fig 5B). Since the angles

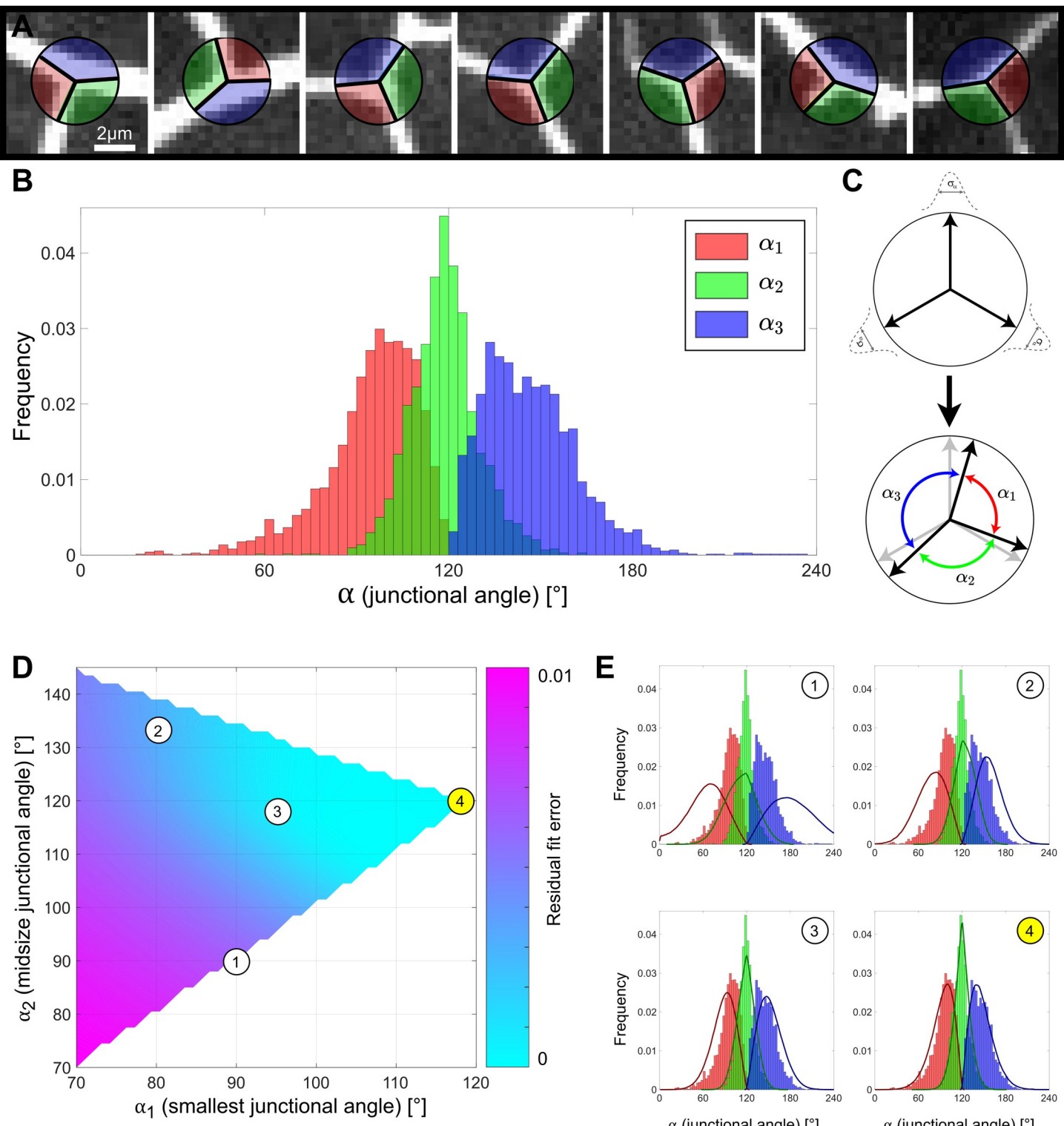

**Fig 5. The geometry of PVD junctions. A.** Examples of various PVD junction morphologies. Colors correspond to relative junctional angle size: smallest (red, $\alpha_1$), midsize (green, $\alpha_2$) and largest (blue, $\alpha_3$). **B.** Each junction is described by three angles, corrected for distortion due to projection. The distributions of the smallest (red), midsize (green) and largest (blue) angles are shown (n = 10 animals; 2620 junctions). The mean values of the angle distributions are 92˚, 119˚ and 149˚. **C.** A cartoon showing the effect of random variations in junction configuration. An intrinsic junction configuration (top) is deformed, resulting in one of the possible deformed configurations (bottom). **D.** Fit of Monte Carlo simulated angle distributions with the experimental distributions. The best fit to the experimental data is for a symmetrical configuration

with a standard deviation of $\sigma_\alpha = 19°$ around the mean values ({$\alpha_1 = 120°$, $\alpha_2 = 120°$, $\alpha_3 = 120°$, $\sigma_\alpha = 19°$}; configuration (4)). Circled numbers correspond to the simulation examples in (E). **E.** Examples of simulated junction configurations. Circled numbers correspond to the ones in (D) to show the residual error for each simulation. Solid lines indicate probability densities for simulated junctions, bars indicate frequencies from experimental data. (1) = {$\alpha_1 = 90°$, $\alpha_2 = 90°$, $\alpha_3 = 180°$, $\sigma_\alpha = 26°$}, (2) = {$\alpha_1 = 80°$, $\alpha_2 = 130°$, $\alpha_3 = 150°$, $\sigma_\alpha = 17°$}, (3) = {$\alpha_1 = 94°$, $\alpha_2 = 118°$, $\alpha_3 = 148°$, $\sigma_\alpha = 15°$}, (4) = {$\alpha_1 = 120°$, $\alpha_2 = 120°$, $\alpha_3 = 120°$, $\sigma_\alpha = 19°$}. Configuration (4) gives the best fit as shown in (D).

in a triplet are not independent, the means of the angle distributions do not correspond to the most abundant junction geometry. In order to reconstruct the charactersitic PVD junction, we used a Monte Carlo algorithm to simulate a range of possible junction geometries under deformation by random noise (see Figs 5C and S5B and S1 Text for details). In order to find the angle triplet and the angle variation that best characterize PVD junctions, we fitted the angle distributions of the simulated junctions with the experimentally determined distributions (Fig 5D and 5E; nondimensional fit residuals shown as a function of only $\alpha_1$ and $\alpha_2$, since $\alpha_3 = 360°$ $-\alpha_1-\alpha_2$). Angle triplets with lower residual errors indicate a better match with the experimental distributions. Importantly, our analysis shows that the experimental data is in best agreement with junction configurations close to the symmetric $\alpha_1 = \alpha_2 = \alpha_3 = 120°$ shape (Fig 5D). Moreover, the fit with the simulation reveals that the PVD junctions are somewhat varied: we found that junction angles are distributed around the symmetric configuration with a standard deviation of $\sigma_\alpha = 18°$ (see S5B Fig; showing the fitted angle standard deviation, $\sigma_\alpha$, as a function of $\alpha_1$ and $\alpha_2$). While this variability of the junction angles allows for intrinsic junction configurations that may deviate from perfect three-way symmetry, these results demonstrate that the orthogonal arrangement of the PVD dendritic arbors (Figs 1C and 4A) is not a consequence of boundary conditions imposed by PVD junctions, but is rather determined by factors that align the processes themselves. The variability of junction geometry further indicates some mechanical flexibility of the junction structure. These findings may be used to guide future works in the search for the physical mechanisms that shape dendritic junctions.

## Distribution of network elements in wild-type and in *git-1* mutant

We next applied our quantitative morphological approach in order to characterize the role of GIT-1, a known regulator of dendritic spines, in shaping the PVD morphology. In order to quantify the effect of GIT-1, we extracted and analysed the PVD dendritic arbor for *git-1 (ok1848)* mutant worms, utilizing the algorithm described for wild-type (WT) worms above (Figs 6A and S6A–S6I). As the most elementary metric for comparison, we quantified the length of dendritic processes detected in each worm. We found that the average total dendritic length was not affected in the mutant, with $6100\pm330\mu m$ (n = 10), compared to $6000\pm270\mu m$ in WT worms (n = 10). Next, we determined how PVD dendrites are distributed into the previously defined morphological classes, using the intrinsic class definitions determined for WT animals. We found that while the total length of PVD dendrites remain unchanged, its distribution into the classes was altered: the combined length of class-4 branches was decreased in the *git-1* mutant compared to WT, whereas the length of class-3 was increased (see Fig 6B). Conversely, we found that the lengths of classes 1 and 2 were not changed considerably.

Next, we investigated the effect of *git-1* mutation on the degree of arborization of the PVD dendrites. We define the one-dimensional junction density, $\rho$, as the number of three-way-junctions per unit length of the dendrites. The junction density is the reciprocal of the mean dendritic distance between neighboring junctions, and therefore constitutes a fundamental structural parameter of the PVD. We found that the overall density of junctions in the PVD of WT animals is $0.042\pm0.003\mu m^{-1}$, and is unchanged in *git-1* mutant worms. Nevertheless, a more detailed analysis revealed that the junction density varies throughout the PVD, with

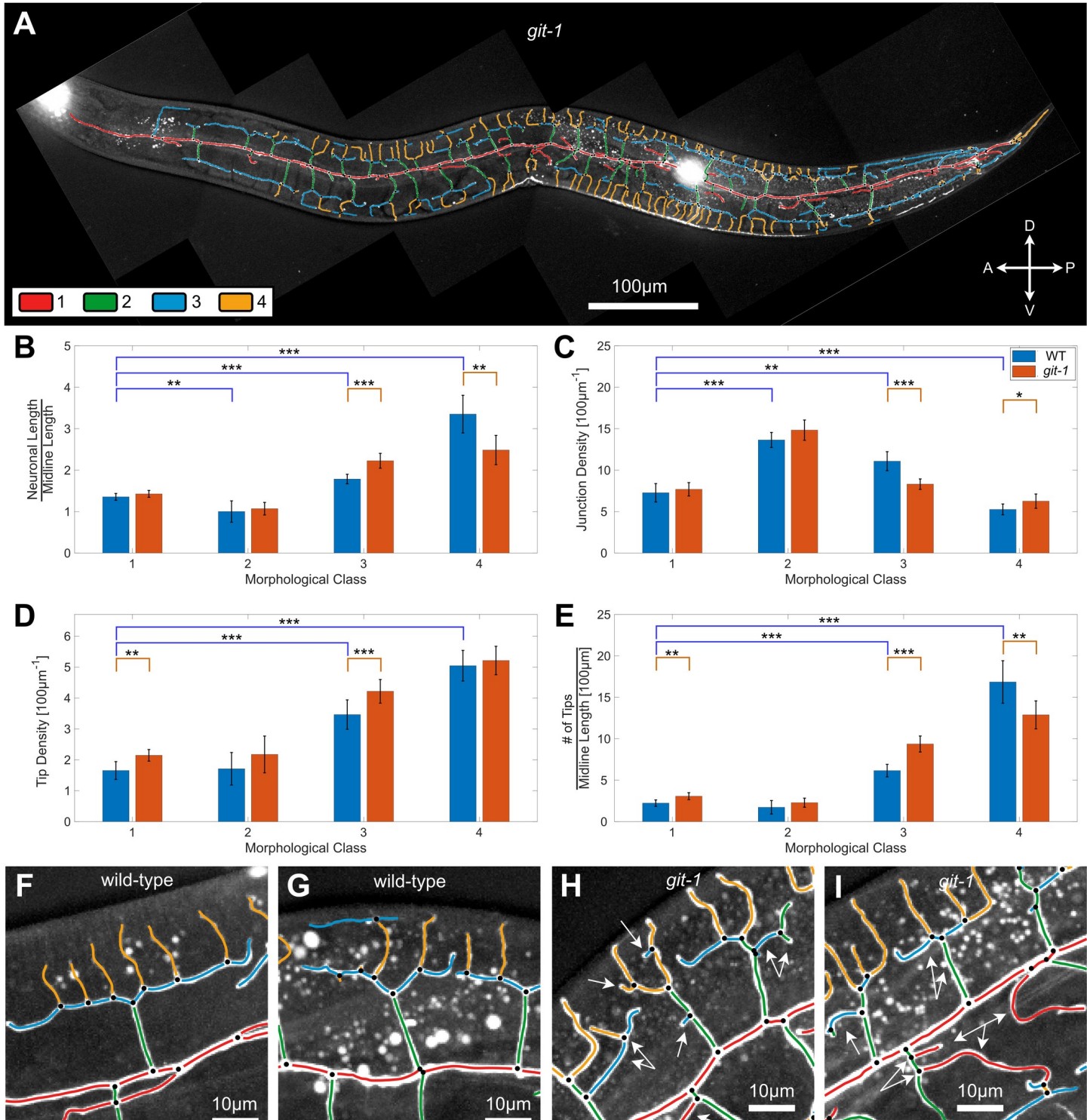

**Fig 6. Distributions of geometrical PVD elements in wild-type and *git-1* mutant. A.** An image of a full PVD of a *git-1* mutant, superimposed with color-coded morphological classes: class 1 (red), class 2 (green), class 3 (blue) and class 4 (yellow). **B.** Total PVD length for each morphological class for WT (blue) and *git-1* mutant (red), normalized to the whole PVD midline length. **C.** The length density of junctions along the dendritic processes (reciprocal of the average length of dendrites between junctions normalized per 100 μm), for each morphological class. **D.** The length density of tips along the dendritic processes, normalized per 100μm, for each morphological class. **E.** The number of dendritic tips, normalized per 100μm of midline length, for each of the four morphological classes. **F-I.** Magnified regions of typical PVD morphologies in wild-type (F-G) and *git-1* (H-I). Arrows show abnormal branching in *git-1* compared to wild-type. This includes excess of junctions and tips, as well

as neuronal processes that do not follow the pattern described in Figs 3 and 4. Dendritic segments color-coded according to classification as in panel A. In B-E, statistics were calculated using the nonparametric Mann–Whitney test. ***$p < 0.0005$, **$p < 0.005$, *$p < 0.05$. n = 10 wild-type animals, with 2620 junctions and 2228 tips. n = 10 *git-1* animals, with 2631 junctions and 2308 tips. Bars show the mean value and error bars show the standard deviation.

significantly elevated junction densities found in the 2nd class, compared to the 1st and 4th classes (see Fig 6C). This trend was observed both in wild-type and *git-1* mutants; however, our analysis has revealed a significant decrease in junction density of class 3 branches in the *git-1* mutant, as compared to WT (Fig 6C).

In addition to 3-way junctions and linear segments, the PVD architecture includes dendritic tips which terminate PVD processes. Such tips may represent sites at which applied mechanical forces can promote dendritic elongation or retraction, as well as potential sites of membrane fusion, leading to the formation of new 3-way junctions. We therefore sought to characterize the effect of *git-1* on the number and distribution of terminal PVD branches. We found that the average number of tips in a wild-type PVD is 220±20 (n = 10 worms) and is essentially unchanged in *git-1* mutants, with 230±20 (n = 10 worms). Similar to junction density, we characterized the density of terminal tips per unit length of PVD for branches of different classes. We found that the tip density in WT worms increases with class number, with the highest tip density found at 4th class (Fig 6D). This result is remarkable in contrast with the junction density, which is lowest at 4th class branches. In *git-1* mutants both tip number and tip density of class 1 and 3 are increased compared to WT, while the number, but not density, of class-4 tips is reduced (Fig 6D and 6E). We note that while the reduction in class-4 tip number and length maintains a tip density similar to WT, this is not the case for class-3, where tip density is increased despite the increase in class-3 length (Fig 6B, 6D and 6E).

While some, but not all, features of the WT and *git-1* morphology can be confirmed by visual inspection and manual quantitative analyses (Fig 6F–6I; see S1 Text), the reliable detection of such nuanced differences demonstrates the power of our quantitative automated and objective approach. Taken together, these findings indicate a subtle but significant change in the morphology of the PVD in *git-1* worms that is most prominent in tertiary-classed branches.

## Behavioral phenotypes of *git-1* mutants may correlate to PVD structure

The PVD neuron is involved in several quantifiable behavioral outputs, among which is the escape response to noxious mechanical stimuli (harsh touch) [19,56,57] and altered crawling waveform [18,57]. Chemical ablation of the entire PVD also made animals slightly slower [18]. Since the structural alterations in the PVD dendrites of *git-1* mutants are subtle, yet significant, we sought to determine whether any of the associated behavioral outputs is similarly altered.

By automated extraction of locomotion parameters from freely-crawling worms, we determine that *git-1* animals display altered crawling gait, consistent for two independent mutant alleles (Fig 7A–7C), Specifically, *git-1* animals show increased crawling wavelength (Fig 7D) and increased speed (Fig 7E). The response of *git-1* mutants to harsh touch, however, remained wild-type-like for both mutant alleles examined (Fig 7F). Thus, gait and proprioception, but not response to noxious mechanical stimuli, are affected in *git-1* mutants, suggesting a correlation between the structural changes in PVD arborization and behavior.

## Discussion

In this work we presented a quantitative approach for morphological characterization of the PVD neuron in *C. elegans*. By combining statistical and rule-based analysis, we were able to automatically segment microscopy images and overcome the noise and artifacts that are

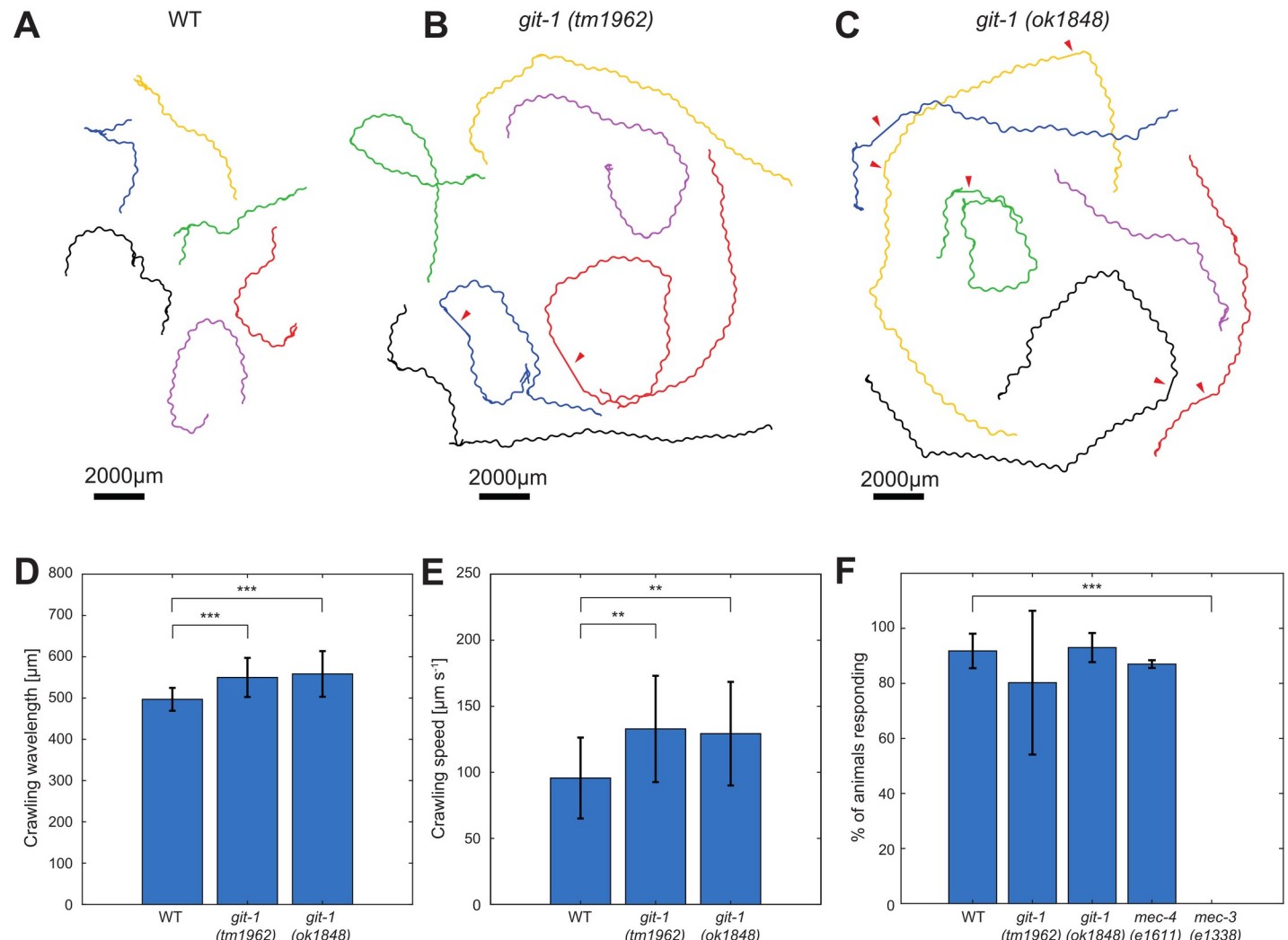

**Fig 7. Behavioral characterization of *git-1* mutants. A-C.** Characteristic tracks obtained by automated video analysis of six animals over three minutes, recorded at 7.5 frames per second (fps), for WT, *git-1(tm1962)* and *git-1(ok1848)* animals, respectively. For each genotype, the animals were recorded on a single plate, and the obtained tracks were separated for clarity. Red arrowheads indicate events where the track was lost (see Materials and Methods). **D.** Crawling wavelength automatically extracted and averaged over a three-minute, 7.5 fps recording. n = 31 WT; n = 15 *git-1(tm1962)*; n = 33 *git-1(ok1848)*. Student's t test, ***$p < 0.0005$. Error bars represent the standard deviation. Values remain significant when normalized per worm length (results not shown). **E.** Crawling speed normalized by total worm length, automatically extracted and averaged over a three-minute, 7.5 fps recording. n = 31 WT; n = 15 *git-1(tm1962)*; n = 33 *git1(ok1848)*. Student's t test, **$p < 0.005$. Error bars represent the standard deviation. Values remain significant when normalized per worm length (results not shown). **F.** Percentage of worms executing a harsh-touch-induced escape response. Results are not significant ($p > 0.01$) compared to WT for all genotypes except the harsh-touch insensitive *mec-3(e1338)*. As a negative control, the gentle-touch insensitive *mec-4(e1611)* animals are used. n = 310 WT; n = 312 *git-1(tm1962)*; n = 250 *git-1(ok1848)*; n = 150 *mec-4(e1611)*; n = 70 *mec-3(e1338)*, n = 4 independent experiments for WT and *git-1(tm1962)* mutants, n = 3 independent experiments for *git-1(ok1848)* mutants and n = 2 independent experiments for *mec-4* and *mec-3* mutants. Fisher's exact test, ***$p < 0.0001$. Error bars represent the standard deviation.

inherent to imaging of dendritic arbors. Following image segmentation, our approach enabled the detection and abstraction of the stereotypical orthogonal patterning of PVD processes.

## Algorithmic classification of dendritic trees

Our analysis produced a bias-free classification of dendritic classes that corresponds well with the established order classification of PVD branches–yet is not identical to it. The algorithmic classification takes into account position and orientation of dendritic elements, but does not

impose other rules, such as connectivity, between branches. Manual classification, on the other hand, is based on a subjective comparison between PVD branches and an idealized menorah structure (Fig 1C). Further, our algorithm classifies all PVD elements according to the closest morphological class, and does not consider indeterminate or mixed cases. Conversely, manual annotations include an idiosyncratic "ectopic" qualifier for branch orders, describing a lack of fit with the idealized menorah shape. The tracing algorithm has been utilized to analyze both wild-type animals as well as mutants with altered PVD geometries. Moreover, we have further tested the algorithm by analyzing the PVD morphology of animals at early developmental stages and found that L4 larvae had fewer branches than young adult animals (S7 Fig). The successul application of the tracing and analysis algorithm to multiple and diverse datasets demonstrates the robustness and generality of our approach.

## Three-way symmetry of PVD junctions generates orthogonal trees

Beyond the ability to automate the inspection of neuronal architecture, our quantitative analysis has uncovered key morphological features that had not been previously detected. We demonstrated that processes are predominantly connected by junctions with a 3-fold symmetry at the branching points. In comparison, studies of human pyramidal neurons reveal that lower branching orders are typically connected by asymmetric junctions, with minimal branching angles close to 60°, which decrease further in higher branching orders [58]. Since PVD processes are, on average, aligned orthogonally in a crosshatch pattern, this unexpected result indicates that the orientation of dendritic processes does not arise from the boundary conditions applied at the junctions, but is rather achieved by factors that act to guide along the length of the processes [23,45,59–63]. Further, our findings imply that symmetrical 3-way junctions represent the mechanically stable junction configuration. This result is in agreement with earlier works that have shown that 3-way junctions represent the minimal energy shape for connecting tubular membrane surfaces [64]. Yet, the observed junction geometry may also be influenced by a local effect of cytoskeletal proteins or membrane coating proteins [65–70]. Our findings can therefore serve as a basis for further study of the physical mechanism for stabilization of PVD junctions.

## PVD junctions are heterogeneously distributed

Our analysis has revealed that the junctions and tips of dendritic processes are not uniformly distributed throughout the PVD. Class-4 branches have the highest number of tips per dendritic length, as might be expected, but also the lowest number of dendritic junctions per unit length. Conversely, class-2 branches have the highest junction density and the sparsest tip density. How may these findings be understood? Formation of a dendritic spine from an existing process results in an addition of one junction and one tip to the network. Therefore, modulating the process of spine formation is not expected to result in a disparity between the junction and tip densities. However, fusion of a dendritic tip will increase the junction density and reduce the density of tips, thereby enhancing the tip-junction disparity. These results therefore suggest that dendritic fusion is enhanced near the primary PVD process and diminishes further away from it. The functions of the fusogen EFF-1 in the dynamics of dendritic architecture and maintenance of the trees may explain this phenomenon [17,71]. Thus, EFF-1-dependent loop formation could be part of the molecular machinery that controls the number of dendritic tips and three-way junctions. Further work is required for detailed analysis of dendritic loops and for determining the factors governing the PVD topology.

## Mutations in *git-1* are associated with behavioral changes mediated in part by the PVD

*git-1* in *C. elegans* has been linked to transduction of muscle tension signals in the elongating embryo [72], however this form of mechanotransduction is different from the ion-channel-mediated sensory input related to PVD activation [19,57,73]. As an example, the mechanosensory channel component *mec-10* is required for PVD activation following harsh touch [19]. *mec-10* mutants show significantly reduced crawling wavelength and amplitude, which were rescued by PVD-specific MEC-10 expression [57,73]. Despite this proprioceptive phenotype, *mec-10* mutants remain wild-type in their harsh-touch response [57].

Our behavioral results support this separation between harsh-touch responsiveness and proprioception. While the alterations to crawling speed and wavelength found here cannot be directly associated with the PVD, they represent a consistent behavioral phenotype of *git-1* mutation to a PVD-related modality. This phenotype is independent from nociception, and is not entirely aligned with known effects of PVD chemical ablation [18] or sensory channel mutations [57].

## Mutation in *git-1* results in class-specific deformation of PVD geometry

We have found that *git-1* mutation results in a subtle deformation of the PVD architecture. No significant differences were detected for the curvature, or the relative distribution into morphological classes. However, we found a significant decrease in the density of junctions along class-3 dendritic process in *git-1* mutant worms, relative to WT. Conversely, the density of class-3 tips was found to increase in the mutant, suggesting that *git-1* acts as a negative regulator of tertiary branch arborization. These findings were confirmed by manual analysis, and we conclude that *git-1* is necessary for both morphogenesis and maintenance of dendritic structure in adulthood. The behavioral phenotypes associated with *git-1* were tested by locomotion analysis and a harsh touch response assay, where we found that two *git-1* mutant alleles displayed increased speed and body wavelength, however this was not correlated with any pronounced mechanosensory defect (Fig 7). These results correspond to a proposed polymodal nature of the PVD, wherein proprioceptive component may be separate from nociception [57].

Previous studies have shown that the PVD dendrite develops beneath the skin-like hypodermal tissue, with its terminal processes closely associated with body wall muscles [17,18,63]. This proximity is hypothesized to play a role in PVD sensory functions, such that variation in the spatial distribution of high-order PVD branches may lead to modulation of proprioception. Theoretical models for *C. elegans* crawling motility [74,75], and for locomotion of soft-bodied animals in general [76], have demonstrated that a feedback mechanism between localized sensing of mechanical stress and contraction of muscle cells is critical for crawling locomotion in the absence of a central pattern generator. Moreover, such models have shown that modulation of the proprioceptive response is expected to affect the muscle contraction waveform and, subsequently, the animal's gait [76]. Taken together, our results open a possibility for novel roles for GIT-1 in dendritic arborization and proprioceptive faculties in *C. elegans*.

## Outlook and future directions

The analysis of the PVD morphology was performed on static images, yet afforded clues to dynamical processes that shape the PVD, such as junction formation and tip fusion. However, the autonomous capabilities of the image segmentation and characterization algorithms are uniquely suited for the processing of a time series of images. In future works, we plan to utilize

the tools we have developed in order to study the time dependent development and aging of the dendritic arbor, in order to uncover the mechanisms through which the PVD morphology is developed and maintained [21]. The morphological database that stores the abstracted PVD shapes of individual worms is designed to allow for novel analysis and future expansions. In order to facilitate such studies, we have made the source code of the algorithm, including a custom graphical user interface for easy implementation, available as open code. Finally, while the image segmentation and analysis algorithms have been developed specifically for the PVD neuron, they are based upon low level geometrical features, rather than predefined "platonic" concepts of the ideal PVD morphology. Consequently, we expect the approach presented here to be applicable for segmentation and classification of arbors in other neurons, with different characteristic architectures.

## Materials & methods

### *C. elegans* strains and maintenance

Nematode strains were maintained according to standard methods [77,78]. Strains were maintained on NGM plates with 150 μl OP50 bacteria at 20˚C. N2 (wild-type) and *git-1* animals were transferred into separate plates at the late L4 stage using a dissecting scope ('Christmas tree' stage vulva, appearing at late L4-very early young adult, as confirmed by Nomarski optics), and imaged 24 hours later, being one-day adults. For S7 Fig, BP2117 animals were either imaged at late L4 stage (defined as described above) or as young adults, defined by presence of first 1–10 eggs ordered in a single row, as confirmed by Nomarski optics. In addition to *C. elegans* strain N2 the following strains were used: BP709 *[hmnIs133 (ser-2prom3::kaede)]* [22], RB1540 *git-1(ok1848) X*, BP1054 *git-1(ok1848) X; hmnIs133 (ser-2prom3::kaede)*. BP1077 *git-1(tm1962) X; hmnEx133(ser-2prom3::kaede)*, (a non-mutant *hmnIs133 (ser-2prom3::kaede)* sibling of the cross which derived BP1077 was utilized as WT), BP2117 *dzIs53[pF49H12.4:: mCherry]; hyEx372[pmyo-2::GFP, pdes-2::nhr-25,bluescript]*. *git-1(ok1848)* was generated by the *C. elegans* Gene Knockout Project at the Oklahoma Medical Research Foundation which is part of the International *C. elegans* Gene Knockout Consortium. *git-1(tm1962)* was generated by the National Bioresource Project, Tokyo, Japan, which is part of the International *C. elegans* Gene Knockout Consortium. BP1077 and BP1054 were generated by crossing BP709 with the *git-1* alleles and the genotypes were confirmed by PCR.

### Image acquisition

One-day adult hermaphrodites were anaesthetized for >20 minutes in a drop of 0.01% tetramisole (Sigma T1512) in M9 buffer on a 3% agar pad. Images of the PVD neuron in hermaphrodite *C. elegans* were analyzed by Nomarski optics and fluorescence microscopy using a Nikon eclipse Ti inverted microscope equipped with Yokogawa CSU-X1 spinning disk confocal microscope fitted with a 40 x oil Plan Fluor NA = 1.3 lens [22]. We used projections of z-stacked confocal images taken at ~0.5–0.6 μm intervals, digitally stored using iXon EMCCD camera (Andor). The acquisition of the images was done using MetaMorph software, utilizing 25% laser intensity at 488 nm, gain of 200 and 300 ms exposure for image acquisition. The multi-dimensional data (in the form of z-stacks) was projected onto a 2D plane using maximal intensity projection in Fiji/ImageJ software (NIH) to produce a 2D grayscale image.

For S7 Fig, L4s or young adult BP2117 hermaphrodites in which *pmyo-2::GFP* expression was not observed (and hence the *hyEx372* extrachromosmal array was likely lost) were anesthetized and imaged as described above with the following exceptions: Apochromat 60x NA = 1.4 lens was used with 561 nm wavelength laser excitation (15–20% intensity, 100 ms imaging exposure time).

### Image preprocessing

The complete PVD was pieced together from smaller PVD regions using Adobe Photoshop CS version 8.0 and 5.0 (Adobe Inc.) by utilizing File-Automate-Photomerge option. A black layer was added under the original image in Adobe Photoshop CS5 (Adobe Inc.). In cases where parts of other worms were present in the image, they were removed by covering them with a black polygon.

### Locomotion analysis

Worms were recorded crawling freely on NGM plates which were freshly seeded with a circular patch of 120μl OP50 bacteria in LB medium and left to air dry for 4–6 hours to a matte finish. 6–8 young adult worms from each genotype were then placed using an eyelash on a plate and given 5 minutes to recover from the transfer. The plates were placed on an illuminator table (MBF Bioscience) and recorded for 10–12 minutes at 7.5 frames/second with a resolution of 2592x1944 pixels using an overhanging fixed lens (Nikon AF Micro-Nikkor 60 mm f/2.8D) focused at 10 μm/pixel.

From each video, a section of three minutes was subsequently analyzed using the WormLab software (MBF Bioscience, Williston, VT, USA) to recover average values of relevant parameters such as speed, wavelength and amplitude. Worms were tracked for a minimum of 100 seconds, accounting for short segments of poor tracking, which were manually removed within the software.

### Harsh touch

Adult worms of each genotype grown on an NGM plate containing 100–150μl OP50 were prodded laterally posterior to the vulva by using a platinum wire mounted to a Pasteur pipette ('pick'). A forward acceleration after contact was considered a response. If the worm did not respond, a second contact was made. Animals which did not respond on both contacts were considered non-responsive.

### Statistical analysis

Statistical analysis was performed pairwise using the nonparametric Mann-Whitney U-test (Wilcoxon rank-sum test). P-value significance: ***$p < 0.0005$, ** $p < 0.005$, * $p < 0.05$. 10 animals were used for wild-type, and 10 animals were used for *git-1* mutants. Error bars show the standard deviation.

### Software

The code was implemented in MATLAB R2020b. The complete project code is available on GitHub: https://github.com/Omer1Yuval1/Neuronalyzer.

## Supporting information

**S1 Fig. Neural network architecture for semantic segmentation of images based on SegNet [79].** This architecture is made of an encoder subunit through which the input image shrinks in size, and a decoder subunit through which the image is upsampled back to its original size. The input layer gets 64x64 pixels patches of grayscale neuronal images and applies zero-centering normalization to them. Then, two convolution units (blue squares) are applied, with each unit consisting of a 2D 3x3 convolution layer, followed by a batch normalization layer (BN) and a rectified linear unit (ReLU) layer. This unit is followed by a 2x2 max pooling layer (green square) that reduces the size of the input by a factor of four. This sequence is repeated

three times in the encoder, resulting in a 8x8 feature map with 64 features. The same sequence is repeated in the decoder, but with each max pooling layer replaced by an upsampling layer (purple square) that precedes each sequence of convolutions. Numbers below encoder layers show the output size of the pooling layer, and in the decoder they show the input to the upsampling layer. Finally, the softmax layer (yellow square) takes the output of the last ReLU layer and converts it into a probability distribution that sums up to 1. The last layer of the network is a pixel classification layer (red square). This layer computes the loss (cross-entropy) during training and performs the prediction of one of the predefined classes for new data.
(PDF)

**S2 Fig. The processing of a PVD image through the pipeline (Fig 1D). A.** A maximum intensity, grayscale image of a PVD neuron. Arrows show the anterior (A), posterior (P), dorsal (D) and ventral (V) directions. **B.** The classification of the image into neuron and non-neuron pixels. CNN derived classification appears in red, manually added pixels in blue and manually removed pixels in yellow. **C.** Skeleton image derived from the binary image in B. **D.** The fully traced neuron. **E.** Segmentation of the traced neuron. Different segments appear in different colors.
(PDF)

**S3 Fig. Detection of the neuron midline and borderline used to define the PVD coordinate system. A.** The neuron's trace is used to generate a single blob image. Then, the centerline and boundary of the blob are detected (dashed blue and green lines respectively). These are then refined using a sliding window along the approximated centerline, resulting in the final midline and boundary of the neuron (solid lines). **B.** A PVD color-coded for midline coordinate, from anterior (A) to posterior (P). Each neuron element is associating with a midline point by shortest distance, and colored according to its corresponding midline arclength from anterior to posterior. **C.** The distribution of midline orientation of junction rectangles. This distribution is different from the one of all neuron elements (Fig 3H), with peaks at 13.5˚ and 90˚.
(PDF)

**S4 Fig. Algorithmically-derived morphological classes.** Classes color-coded as in Fig 4A–4E: Class 1 (red), class 2 (green), class 3 (blue) and class 4 (yellow). **A.** Magnified regions of wild-type PVDs algorithmically classified into morphological classes. Arrows show classifications that do not match the conventional manual classification into Menorah orders. **B-J.** Visualization of the algorithmically-derived classification in nine PVD images of wild-type *C. elegans* worms (in addition to the one in Fig 4C).
(PDF)

**S5 Fig. The geometry of PVD junctions. A.** Examples of PVD junction morphologies, showing a two-times larger neighborhood compared with Fig 5A, to demonstrate that process orientation often does not match junctional angles. Colors correspond to relative angle size: smallest (red), mid-size (green) and largest (blue). **B.** The variability in junction geometries is characterized by angular noise, determined from a Monte-Carlo simulation. Fit with simulated distributions gives a junction variability of 18˚ around the symmetrical configuration (120˚-120˚-120˚), as indicated by the star symbol.
(PDF)

**S6 Fig. Algorithmically-derived morphological classes for *git-1(ok1848)* mutants. A-I.** Visualization of the algorithmically-derived classification in nine PVD images of *git-1(ok1848) mutants* (in addition to the one in Fig 6A).
(PDF)

**S7 Fig. Quantification of morphological changes in the PVD during development. A.** An image of a full PVD of an L4 worm, superimposed with color-coded morphological classes: class 1 (red), class 2 (green), class 3 (blue), class 4 (yellow). **B.** An image of a full PVD of a young-adult worm, superimposed with color-coded morphological classes, as in A. **C.** Distribution of neuronal length along the midline (head = 0), averaged across worms, for L4 (blue, n = 5) and young-adult (red, n = 5) worms. **D.** The total PVD length for L4 (blue) and young-adult (red) worms. **E.** The total PVD length for each morphological class for L4 (blue) and young-adult (red) worms. **F.** The total number of dendritic junctions, as described in D. **G.** The total number of dendritic tips, as described in D. **H.** The density of dendritic junctions, as described in D. **I.** The density of dendritic tips, as described in D. **J.** The total number of dendritic junctions for each morphological, as described in E. **K.** The total number of dendritic tips for each morphological, as described in E. In D-K, statistics were calculated using the non-parametric Mann–Whitney test. $^*p < 0.05$. n = 5 L4 animals, with 1218 junctions and 1085 tips. n = 5 young-adult animals, with 1610 junctions and 1410 tips. Bars show the mean value and error bars show the standard deviation.
(PDF)

**S1 Table. Parameter values of the algorithms.**
(DOCX)

**S1 Text. Extended analysis.**
(DOCX)

# Acknowledgments

We thank the Lamm lab for the use of their locomotion imaging system. We thank Sharon Inberg, Tamar Gattegno, Veronika Kravtsov and members of the TS and BP lab for advice and discussions.

# Author Contributions

**Conceptualization:** Benjamin Podbilewicz, Tom Shemesh.

**Investigation:** Omer Yuval, Yael Iosilevskii, Anna Meledin.

**Software:** Omer Yuval.

**Supervision:** Benjamin Podbilewicz, Tom Shemesh.

**Validation:** Yael Iosilevskii.

**Writing – original draft:** Tom Shemesh.

**Writing – review & editing:** Omer Yuval, Yael Iosilevskii, Benjamin Podbilewicz, Tom Shemesh.

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
