## [Decision Letter · Decision Letter 0]

4 Feb 2021

Dear Dr Shemesh,

Thank you very much for submitting your manuscript "Neuron tracing and quantitative analyses of dendritic architecture reveal symmetrical three-way-junctions and phenotypes of git-1 in C. elegans" for consideration at PLOS Computational Biology.

As with all papers reviewed by the journal, your manuscript was reviewed by members of the editorial board and by several independent reviewers. In light of the reviews (below this email), we would like to invite the resubmission of a significantly-revised version that takes into account the reviewers' comments.

We cannot make any decision about publication until we have seen the revised manuscript and your response to the reviewers' comments. Your revised manuscript is also likely to be sent to reviewers for further evaluation.

Sincerely,

Hugues Berry

Associate Editor

PLOS Computational Biology

Jason Haugh

Deputy Editor

PLOS Computational Biology

Reviewer's Responses to Questions

**Comments to the Authors:**

Reviewer #1: In this manuscript, Yuval et al. present a novel approach to study the morphology of dendritic spines, and specifically focus on the C. elegans PVD neuron, which has a quite complex structure. The authors combine a CNN with a tracing algorithm to detect PVD dendrites in a manner much superior than other traditional approaches, such as segmentation and skeletonization. The authors show their system performs very well at tracing this complex neuron. In addition, this work presents quantification metrics that describe the spatial arrangement of dendrites with a mathematical basis. Using this tool, the authors characterize in great length the arrangement of PVD in young animals, and then compare these results to mutants of git-1, which is presented as a regulator for dendritic spines. This work is a very methodical, well-thought approach to PVD characterization, which could be used for analysis of many neuron types. It could certainly be used in many different contexts, to study mutant, disease, or aging phenotypes, and thus shed light on the biological mechanisms that control dendritic structure formation. I believe this article topic and quality is suitable for this journal, after the following few items have been addressed.

1. It is unclear why background subtraction is performed after the active contour model. It would appear that the CNN algorithm already performs this task. Can the authors clarify this point?

2. Are the scores for each rectangle quantified with the binarized mask? And how is the rectangle height determined? (the authors explain how the width is determined but not the height)

3. It appears that manual corrections follow the tracing algorithm, which is then followed by re-training of the CNN. Two points here:

a. How frequent were manual annotations necessary? How many iterations did the system go through?

b. How would this manually corrected traced PVD help retrain the CNN? I could have misunderstood, but it appears the purpose of the CNN is not binarization but denoising? Clarification in the manuscript text would be very helpful.

4. I suggest modifying the diagram in 3D, if possible, it is a bit difficult to understand what the angle is measuring.

5. For class clustering: how was this done? This is particularly important, since this classification is used when comparing to git-1 mutants. The largest difference is seen in an increase in class 3 and decrease in class 4, but it is unclear how these are found.

6. Related to point 5, from image 4A and C, it is unclear to me how “green dendrites” in C could all be at phi=0, if some are clearly 180 degrees away from each other. Could the authors clarify this point? In addition, how is it possible that “green” and “blue” dendrites are located at different azimuthal positions, if these are connected and the “blue” dendrites typically do not project radially.

7. I suggest adding background on phenotypes previously observed on git-1 mutants in C. elegans, are there phenotypic differences already known/studied in this organism?

8. It would be interesting to add some discussion on how robust the system is to other phenotypes. For instance, old animals exhibit drastic disorganization of PVD. Would this algorithm work as is, or would re-fitting be necessary.

Reviewer #2: This paper shows a nice piece of work at the interface between image processing, statistical analysis and biology.

The authors propose an algorithm for detecting dendritic arborization from C elegan images and some statistical tool to analyze its geometrical and topological properties. Some singificatn differences are shwon between wild type and mutant populations. As such, this paper perfectly fits with PLOS Computational Biology aims.

The paper is well written and the work is of quality. However I have a few remarks before this paper can be considered for publications.

1) The image processing pipeline is not fully clear for me. At first the CNN is proposed to denoise the image. It is then followed by a binarization and by rectangles matching.

- Later on , the CNN is refered to as a classification step : denoising or classification ?

- The text also refers to active contour. Where exactly are used active contour ? If the image has already been binarized, active contour seems useless.

- How is made the binarization : thresholding ? (if yes how is estimated the thrershold).

- Figure 1 and the feedback loop are not clear.

- Contrast is enhanced using photoshop, then there is also a manual correction performed.

If the approach is not fully automatic, one can wonder what gain is obtained compared to a manual

delineation of the dentritic tree.

- for denoising, simple filters such a median or anisotropic filtering could be also efficient. Did you try before using CNN ?

- How the CNN was trained ? How many samples in the learning set ?

- How is evaluated the dentritic tree detection/segmentation ? compared with classical approaches ?

2) why monte carlo simulation are needed to study angle distribution ? Gaussian mixture models

seem appropriate.

3) can you elaborate on this assertion "While this variability of the junction angles allows for intrinsic

junction configurations that may deviate from perfect three-way symmetry, these results

demonstrate that the orthogonal arrangement of the PVD dendritic arbors (Fig. 1C and 4A) is

not a consequence of boundary conditions imposed by PVD junctions, but is rather determined

by factors that align the processes themselves"

Reviewer #3: This work gathers the analyses of morphological data of PVD neurons (wild type and GIT-1 mutants) in C. elegans worms. To achieve this goal, the authors use new quantitative techniques that can potentially be of interest to the worm community. The proposed new imaging processing methods may also facilitate morphological data analysis of PVD neurons in future high-throughput developmental studies and genetic screens. The manuscript has good presentation, however, it suffers from a slight lack of focus. It is hard for the reader to understand if this is a tools and resources paper, or a more biological oriented piece. In order for the manuscript to be better received by the community it could benefit from 1) better curated reference list; 2) new data analysis that highlights the proposed methods large-scale applicability; and/or 3) more detailed description and understanding of the structure-function relationship of the PVD neurons wild type and mutants alike. Please see my concerns in more detail below:

Minor concerns:

1) Some of the citations in the introduction concerning the structure-function relationship in dendrites seem to be from a number of tangential studies that were cited just to bulk up the references. I feel that the emphasis should also be put on the vast literature concerning the structure-function relationship in dendrites in nematodes (e.g. C elegans, Drosophila larva). Finally, there are a number of recent studies that focus on quantitative analysis of dendritic structure-function relationship -very relevant to this work- that are not cited in the present manuscript. I believe that the paper will benefit the field more if these changes are accommodated.

Essential/Major concerns:

2) The authors present an interesting image tracing tool and morphological analysis of PVD dendrite structure. While the methods/techniques presented are not entirely novel, they have been optimised for this system which could benefit future studies in the field. If the authors wished for the present manuscript to be a tools and resources paper, rewriting the paper is needed - mostly results and discussion (and Methods in coordination). That is, the paper should focus on the tool and its usage, and not mainly about the subsequent results. The authors need to be clear about what the new image tracing tool brings forth relative to available “off-the-shelf” tools in the field. A title change should be considered in light of these changes. New datasets (e.g., mutants, time-lapse data) should also be analysed to prove the general applicability of the new tracing and analysis methods. These aforementioned datasets could be available to the community already - there is no need to acquire new data. Tutorials on how to use the tool should also be provided, or other technical aspects required for potential users to apply the method on their own data.

3) Even though the analysis of the morphology of PVD cells is sound, it is not extensive - there is a vast literature of dendrite morphometrics to quantify such tree-like structures that were not used. Moreover, the authors put very little emphasis on interpreting the functional consequences of such results. As an example, only in page 17, lines 18-23, the authors discuss/provide possible functional/behavioural correlates of the found morphological differences in PVD neurons GIT-1 mutants. From a biological point of view, the importance of the study is not clear. Why are three-way-junctions important for the PVD sensory role? Is there any loss-of-function in PVD neurons GIT-1 mutant due to localized reduction in junctions? What mechanistic insights of the PVD system does the reader get after reading the paper? In case the authors wish to present this work as a biological oriented paper, perhaps they should consider expanding on the behavioral phenotypes of GIT-1 and contrast the structure-function relationship of PVD cells wild type vs GIT-1.

Concluding, essential concerns 2) and 3) need to be addressed. If the authors wish to rewrite the manuscript to focus more on the tools, then essential concern 2) needs to be extensively addressed. If that is not the case, and the authors prefer to provide a deeper analysis of PVD neurons structure-function relationship, then the points raised in essential concern 3) have to be explored. Ideally, both essential concerns would be addressed, if the authors wish to do so. Finally, the link to the computer code is not available:

https://github.com/Omer1Yuval1/Neuronalyzer

**Have all data underlying the figures and results presented in the manuscript been provided?**

Reviewer #1: Yes

Reviewer #2: Yes

Reviewer #3: **No: **The link to the computer code and data is not available:

https://github.com/Omer1Yuval1/Neuronalyzer

PLOS authors have the option to publish the peer review history of their article (what does this mean?). If published, this will include your full peer review and any attached files.

Reviewer #1: No

Reviewer #2: **Yes: **Xavier Descombes

Reviewer #3: No
---

## [Decision Letter · Decision Letter 1]

28 May 2021

Dear Shemesh,

Thank you very much for submitting your manuscript "Neuron tracing and quantitative analyses of dendritic architecture reveal symmetrical three-way-junctions and phenotypes of git-1 in C. elegans" for consideration at PLOS Computational Biology. As with all papers reviewed by the journal, your manuscript was reviewed by members of the editorial board and by several independent reviewers. The reviewers appreciated the attention to an important topic. Based on the reviews, we are likely to accept this manuscript for publication, providing that you modify the manuscript according to the recommendation of reviewer #3.

Sincerely,

Hugues Berry

Associate Editor

PLOS Computational Biology

Jason Haugh

Deputy Editor

PLOS Computational Biology

[LINK]

Reviewer's Responses to Questions

**Comments to the Authors:**

Reviewer #1: In this revised version, Yuval et al have adequately addressed the raised points in the prior submission. The role of the CNN has been made clearer, and the inclusion of additional characterization of git-1 mutants corroborates the importance of assessing neuron structure. The authors have carefully considered all the raised points and present a manuscript where the image processing and analysis of PVD structure is clear.

Reviewer #2: My comments have been successfully adressed. The manuscript

is now much clearer.

Reviewer #3: The authors addressed the majority of the points previously raised. The reference list was updated and new datasets were used to highlight the applicability of the proposed methods to other datasets. Additionally, the authors included results for motility analysis and harsh-touch experiments of git-1 mutant animals. This new analysis greatly increased the reader’s understanding of the structure-function relationship of PVD neurons. Overall, the readability of the manuscript improved substantially. However, a more detailed description of the a link between the structural and functional phenotypes is still missing. Please see my concern in more detail below:

Minor concerns:

1. From a biological point of view, a putative mechanism linking PVD git-1 mutants structural and functional phenotypes is still missing. In case the authors wish to do so, they could consider speculate -based on previous literature- how the observed structural changes in the morphology of the PVD in git-1 worms relates to the observed behavioural phenotypes.

**Have the authors made all data and (if applicable) computational code underlying the findings in their manuscript fully available?**

Reviewer #1: **No: **I might have missed it, but I could not find where image data would be located for sharing

Reviewer #2: None

Reviewer #3: Yes

PLOS authors have the option to publish the peer review history of their article (what does this mean?). If published, this will include your full peer review and any attached files.

Reviewer #1: No

Reviewer #2: No

Reviewer #3: No

Figure Files:

Data Requirements:

Reproducibility:

References:

---

## [Editor Report · Decision Letter 2]

15 Jun 2021

Dear Shemesh,

We are pleased to inform you that your manuscript 'Neuron tracing and quantitative analyses of dendritic architecture reveal symmetrical three-way-junctions and phenotypes of git-1 in C. elegans' has been provisionally accepted for publication in PLOS Computational Biology.

Best regards,

Hugues Berry

Associate Editor

PLOS Computational Biology

Jason Haugh

Deputy Editor

PLOS Computational Biology

---

## [Editor Report · Acceptance letter]

9 Jul 2021

PCOMPBIOL-D-20-02218R2 

Neuron tracing and quantitative analyses of dendritic architecture reveal symmetrical three-way-junctions and phenotypes of git-1 in C. elegans

Dear Dr Shemesh,

I am pleased to inform you that your manuscript has been formally accepted for publication in PLOS Computational Biology. Your manuscript is now with our production department and you will be notified of the publication date in due course.

With kind regards,

Katalin Szabo
